# Generation of Stable Cell Lines Expressing Akabane Virus N Protein and Insight into Its Function in Viral Replication

**DOI:** 10.3390/pathogens12081058

**Published:** 2023-08-18

**Authors:** Jingjing Wang, Dongjie Chen, Fang Wei, Junhua Deng, Jia Su, Xiangmei Lin, Shaoqiang Wu

**Affiliations:** 1Institute of Animal Inspection and Quarantine, Chinese Academy of Inspection and Quarantine, Beijing 100176, China; jing_j_w@163.com (J.W.); chendongjie@caiq.org.cn (D.C.); weif@caiq.org.cn (F.W.); dengjh@caiq.org.cn (J.D.); linxm@caiq.org.cn (X.L.); 2China Institute of Veterinary Drug Control, Beijing 100081, China; hnzk912@163.com

**Keywords:** Akabane virus (AKAV), nucleocapsid (N) protein, stable BHK-21 cells, inhibition, viral replication, mRNA expression

## Abstract

Akabane virus (AKAV) is a world wide epidemic arbovirus belonging to the *Bunyavirales* order that predominantly infects livestock and causes severe congenital malformations. The nucleocapsid (N) protein of AKAV possesses multiple important functions in the virus life cycle, and it is an ideal choice for AKAV detection. In this study, we successfully constructed two stable BHK-21 cell lines (C8H2 and F7E5) that constitutively express the AKAV N protein using a lentivirus system combined with puromycin selection. RT-PCR analysis confirmed that the AKAV N gene was integrated into the BHK-21 cell genome and consistently transcribed. Indirect immunofluorescence (IFA) and Western blot (WB) assays proved that both C8H2 and F7E5 cells could react with the AKAV N protein mAb specifically, indicating potential applications in AKAV detection. Furthermore, we analyzed the growth kinetics of AKAV in the C8H2 and F7E5 cell lines and observed temporary inhibition of viral replication at 12, 24 and 36 h postinfection (hpi) compared to BHK-21 cells. Subsequent investigations suggested that the reduced viral replication was linked to the down-regulation of the viral mRNAs (Gc and RdRp). In summary, we have established materials for detecting AKAV and gained new insights into the function of the AKAV N protein.

## 1. Introduction

Akabane disease mainly causes reproductive losses in pregnant ruminants, including cattle, sheep, and goats, which are characterized by abortion, stillbirth, premature birth, and congenital deformities [1]. As an arthropod-borne disease that is mainly transmitted by Culicoides biting midges [1], outbreaks of Akabane diseases have been reported in various regions worldwide, including Australia, Asia, the Middle East, and Africa [2,3,4]. In the 1970s, the Japanese livestock industry suffered significant economic losses due to outbreaks of Akabane disease [4,5]. In recent years, the northeast region of China has experienced frequent epidemics of this disease, leading to substantial economic losses in the cattle industry.

Akabane virus (AKAV), the causative agent of Akabane disease, was first isolated and identified in Japan in 1959 [6] and was classified into the Simbu serogroup, genus *Orthobunyavirus*, family *Peribunyaviridae* and order *Bunyavirales* [7,8]. Similar to that of other orthobunyaviruses, the genome of AKAV consists of three segments of single-stranded negative-sense RNA, denoted as large (L), medium (M), and small (S) segments. The L segment encodes the RNA-dependent RNA polymerase (RdRp), whereas the M segment encodes two enveloped glycoproteins (Gn and Gc) and a non-structural protein, NSm. The S segment, the most conserved one, encodes for a nucleocapsid protein N and a non-structural protein NSs [9,10]. N protein is a group reactive antigen that can elicit a strong humoral immune response by viruses belonging to the serogroup [11,12,13,14,15,16] and its conserved antigenicity has been proven in many reassortants [10,17]. After infection, the N protein is abundantly produced in infected cells, and anti-N antibodies are highly abundant in infected animals [18,19]. These properties enable the N protein to be a promising target for AKAV detection. The N protein is also essential for RNA replication and transcription, as it interacts with viral RNA species to form ribonucleoprotein complexes (RNPs) [20].

This study involved the creation and characterization of stable BHK-21 cell lines that constantly produce the AKAV N protein. Additionally, we also investigated the growth kinetics of AKAV in these cell lines and found a temporary inhibition of the viral replication efficiency.

## 2. Materials and Methods

### 2.1. Cell Lines, Cell Culture and Viruses

293T cells and BHK-21 cells were stored in our laboratory (CAIQ, the Institute of Animal Inspection and Quarantine, Chinese Academy of Inspection and Quarantine), and cultured in Dulbecco’s modified Eagle’s medium (DMEM; Gibco) supplemented with 10% fetal bovine serum (FBS; Gibco). AKAV insolate TJ2016 (GenBank accession no. MT761689, MT761688 and MT755621) used in this study was isolated from the serum of bovine and identified and stored by our lab [21].

### 2.2. Antibodies and Plasmids

The monoclonal antibody (mAb) 2D3 that was raised against the AKAV N protein [22] was prepared in our laboratory. Mouse anti-β-actin mAb, FITC-labeled goat anti-mouse IgG, and RBITC-labeled goat anti-mouse IgG were purchased from SolarbioLife Sciences (Beijing, China). The lentivirus packaging system, which includes a transfer vector (pLV-EF1a-EGFP(2A)Puro) and two helper plasmids (pH1 and pH2), was purchased from Inovogen Tech. Co., Ltd., (Chongqing, China). Information about these plasmids is available on the websites http://inovogen.com/lentivirus/lentivirusvector/2019/0531/587.html (accessed on 14 August 2023), and http://inovogen.com/lentivirus/packaging-system/URL (accessed on 14 August 2023). The plasmid TVT7R-AKAV-S, which bears the S segment of AKAV, was constructed in our lab previously [21].

### 2.3. Generation of the Recombinant Plasmid and the Recombinant Lentiviruses

The schematic diagram for the construction of the recombinant plasmid is shown in Figure 1. Briefly, the AKAV N gene was amplified by polymerase chain reaction (PCR) from plasmid TVT7R-AKAV-S with primer set N-F1/N-R1 (Table 1). To assemble the N gene into the vector pLV-EF1a-EGFP(2A)Puro, *Xba*I and *Sma*I were introduced into the 5′ and 3′ ends of the fragment, respectively. The PCR amplicon and pLV-EF1a-EGFP(2A)Puro vector were simultaneously digested with *Xba*I and *Sma*I (TaKaRa Biotechnology) and ligated with T4 ligase (TaKaRa Biotechnology, Kusatsu, Japan). The generated recombinant plasmid was identified by genomic sequencing (TSINGKE Biological Technology, Nanjing, China) and was designated as pLV-EGFP-AKAV-N. The recombinant vector pLV-EGFP-AKAV-N and two helper plasmids (pH1 and pH2) were extracted from Trans10 chemically competent cells (TransGenBiotech, Beijing, China) with the EasyPureHiPure plasmid MaxiPrep kit (TransGen Biotech). The extracted plasmids were assessed for quality and stored at −80 °C until required.

Further, 293T cells were plated into 10cm dishes and incubated at 37 °C in a humidified incubator containing 5% CO_2_ overnight until they reached approximately 90% confluence. The growth medium was removed and replaced with 10 mL of FBS-free DMEM two hours before transfection. Plasmid pLV-EGFP-AKAV-N (5 µg), helper plasmid pH1 (3.75 µg) and helper pH2 (1.25 µg) were co-transfected into 293T cells along with the Lipofectamine LTX and Plus Reagent (Invitrogen, Life Technologies, USA) according to the manufacturer’s protocol. Six hours posttransfection, the mixture was removed and replaced with 10 mL of DMEM containing 10% FBS. The supernatants containing the recovered recombinant lentiviruses were collected at 48 h posttransfection. The harvested supernatants were centrifuged to remove cellular debris, filtered through a 0.22-µm filter and stored in aliquots at −80 °C.

### 2.4. Transduction and Puromycin Selection of BHK-21 Cells

BHK-21 cells were transduced by recombinant lentiviruses according to the methods described previously [23]. Briefly, BHK-21 cells were plated in 24-well plates (Corning, Corning, NY, USA) and grew overnight until the cultures reached about 60% confluence. The cells were rinsed, and each well received 450 µL of cell culture medium, 50 µL of recombinant lentiviruses stocks, and 3 µg of polybrene (Inovogen Tech. Co., Ltd., Chongqing, China). The plate was placed in a 37 °C/5% CO_2_ incubator for 24 h. Then, the media was replaced with DMEM containing 10% FBS, and cells were cultured for an additional 24 h. Forty-eight hours after transduction, the transduced BHK-21 cells were trypsinized and seeded in 6-well plates (Corning, Corning, NY, USA) at a density of 100~200 cells/well, cultured with fresh DMEM containing 10% FBS and 5 µg/mL puromycin (Inovogen Tech. Co., Ltd., Chongqing, China) for selection. The selective medium was exchanged every 3~4 days for up to three weeks until nearly all of the EGFP-negative cells were killed. The surviving cell colonies were screened and sub-cloned at least three times by the limiting dilution method. Finally, two positive BHK-21 cell lines that expressed the N protein of AKAV were established and were designated as C8H2 and F7E5, respectively.

### 2.5. Western Blot

To analyze the levels of AKAV nucleocapsid (N) protein expressed in the established cell lines, the 5th, 15th, and 30th generations of C8H2 and F7E5 cells were cultured in 6-well plates with the same density. Forty-eight hours later, the total protein of each cell line was extracted with RIPA lysis buffer (Beyotime, Shanghai, China) following the manufacturer’s instructions. The samples were separated by 12%SDS-PAGE and transferred onto polyvinylidene fluoride (PVDF) membranes (Millipore, Boston, MA, USA). The PVDF membranes were blocked and probed with primary antibodies. MAb 2D3 (1:10,000 dilution) was used for the detection of the AKAV N protein, and mAb anti-β-actin (1:10,000 dilution) was used as an internal endogenous control. Afterward, the membranes were washed and incubated with the secondary antibody HRP-conjugated goat anti-mouse IgG (1:10,000 dilution). The membranes were further incubated with the EasySee Western blot kit (TransGen Biotech, Beijing, China) and finally exposed to a FluorChem E apparatus (ProteinSimple, Santa Clara, CA, USA).

### 2.6. Immunofluorescence Assay (IFA)

C8H2, F7E5 and BHK-21 cells were seeded into a 24-well plate, respectively, and incubated at 37 °C with 5% CO_2_ overnight. After washing with PBS, the cells were fixed with pre-cold absolute ethyl alcohol for 30 min. MAb 2D3 (1:1000 dilution) was used as the primary antibody, and the RBITC-labeled goat anti-mouse IgG was used as the secondary antibody. Fluorescence images were obtained by the Invitrogen EVOS FL cell fluorescence imaging system (Thermo Fisher Scientific, Waltham, MA, USA).

### 2.7. RT-PCR Assays

To demonstratethe integration of the AKAV N gene in the established cell lines, RNA from the 5th, 15th, and 30th generations of C8H2 and F7E5 cells was extracted using the RNA Easy Fast Tissue/Cell Kit (TianGen Biotech, Beijing, China) according to the manufacturer’s instructions and then treated with RNase-free DNaseI (TianGen Biotech, Beijing, China) to remove the residual genomic DNA. The extracted RNA was immediately reverse-transcribed into full-length cDNAs with the FastKing RT kit (with gDNase) (TianGen Biotech, Beijing, China), which served as the amplification template for the next step. The cDNA of BHK-21 cells and AKAV-infected BHK-21 cells were used as the negative control and positive control, respectively. The AKAV N gene was detected by PCR with primer set N-F2/N-R2 (Table 1). The PCR products were sent to Tsingke Biological Technology for genomic sequencing.

### 2.8. qRT-PCR Assays

To evaluate the relative mRNA levels of the AKAV gene in the indicated cell lines, total RNA from the cells was extracted and reverse-transcribed into full-length cDNAs according to the method described in Section 2.7. The qRT-PCR assays were conducted with SYBR green master mix (Vazyme Biotech, Nanjing) on an iCycler iQ5™ System (Bio-Rad, Hercules, CA, USA) according to the manufacturer’s instructions. The Gc and RdRp genes of AKAV were amplified with primer sets Gc-qF/Gc-qR and RdRp-qF/RdRp-qR, respectively (Table 1). The GADPH gene was simultaneously detected with primer set GADPH-qF/GADPH-qR (Table 1) as an internal endogenous control.

### 2.9. Growth Kinetics of AKAV in Different Cell Lines

To evaluate the growth characteristics of AKAV in the established cell lines, C8H2 cells, F7E5 cells, and BHK-21 cells were seeded into 24-well plates and inoculated with AKAV at an MOI of 0.01, respectively. Virus titers at different time points, including 0, 12, 24, 36, 48, 60, and 72 h postinfection (hpi), were determined by using a microtitration infectivity assay in BHK-21 cells. Briefly, confluent monolayers of BHK-21 cells cultured in 96-well plates were inoculated with ten fold serially diluted virus suspensions. After absorption for 2 h in a 37 °C incubator, the inoculum was removed and replaced with fresh DMEM supplemented with 2% FBS. The plates were incubated for an additional 48 h, followed by the implementation of the IFA assay as described above. MAb 2D3 was used as the primary antibody, and FITC-labeled goat anti-mouse IgG was used as the secondary antibody. Virus titers were determined and recorded as 50% tissue culture infective doses (TCID_50_) using the Reed–Muench method.

### 2.10. The Sensitivity of Sera Detection with Purified AKAV N Protein

AKAV N protein was extracted from C8H2 and F7E5 cells and purified using an immunomagnetic bead-based method with rProtein A/G MagPoly Beads (IP Grade) following the instructions provided in the kit manual. The purified N protein was analyzed by SDS-PAGE, and the protein concentration was determined by the BCA assay.

Then, 96-well ELISA plates were coated with 200 ng of purified AKAV-N protein and incubated at 4 °C overnight. The plate was washed with PBST three times and blocked with 4%BSA-PBS at 37 °C for 2 h. After washing three times with PBST, 100µL/well of AKAV antiserum or negative control serum diluted at a ratio of 2^1^~2^15^ fold were added into the wells in triplicate. After 1 h of incubation at 37 °C, the plate was washed with PBST three times again. HRP-labeled rabbit anti-bovine IgG was added to each well and incubated at 37 °C for another 1 h. After washing, TMB substrate solution was added into each well for 15 min, and the reaction was stopped by 2 N H_2_SO_4_. The absorbance was measured at 450 nm using a Multiskan spectrum (Thermo Fisher Scientific, Saint Louis, MO, USA).

### 2.11. Statistical Analysis

Statistical analysis was performed using GraphPad Prism 5 (GraphPad Software Inc., San Diego, CA, USA). Intergroup differences were analyzed using a two-way analysis of variance (ANOVA) followed by a Bonferroni posttest. Values were considered statistically significant if *p* < 0.05 and extremely significant if *p* < 0.01 or <0.001.

## 3. Results

### 3.1. Establishment of Stable BHK-21 Cell Lines for Constitutively Expressing the AKAV N Protein

The constructed recombinant plasmid was confirmed by sequencing. The results showed that the full-length AKAV N gene coding sequence was successfully inserted into the lentivirus transfer vector pLV-EF1a-EGFP(2A)Puro between *Xba*I and *Sma*I (Figure 2). The successfully constructed plasmid was designated as pLV-EGFP-AKAV-N.

Plasmid pLV-EGFP-AKAV-N together with two helper plasmids, pH1 and pH2, were co-transfected into 293T cells, and the expression of EGFP was observed under a fluorescence microscope after 24 and 48 h posttransfection (Figure 3A). When the transfection rate reached more than 95%, the recombinant lentiviruses were harvested and transduced into BHK-21 cells. After puromycin selection and sub-cloning, two positive BHK-21 cell lines that stably expressed EGFP were obtained and were designated as C8H2 and F7E5, respectively. As shown in Figure 3B, all of these cell lines emit a strong green fluorescent signal throughout the cytoplasm.

### 3.2. Characterization of the Established Cell Lines

The levels of AKAV N protein in the C8H2 and F7E5 cell lines were analyzed by western blot (WB) and IFA assays. As shown in Figure 4A, a single protein band was detected by 2D3 mAb in the lysate of each generation of C8H2 and F7E5 cells. For IFA analysis, 2D3 mAb was used as the primary antibody, and a RBITC-labeled goat anti-mouse IgG was used as the secondary antibody. The result of IFA showed that the red immunostaining signals corresponding to the AKAV N protein in C8H2 and F7E5 cells were distributed throughout the cytoplasm (Figure 4B). WB and IFA results indicated that the AKAV N protein was successfully expressed in the C8H2 and F7E5 cell lines.

To demonstratethe integration of the AKAV N gene in the established cell lines, RNA from the 5th, 15th, and 30th generations of C8H2 and F7E5 cells was extracted, reverse transcribed, and detected by PCR assays. As Figure 5 shows, positive bands with 649 bp were detected in all the tested samples. Then, the PCR products were sequenced, and no mutation was found. The result demonstrated that the N gene of AKAV has been stably integrated into the genomes of C8H2 and F7E5 cell lines.

### 3.3. The Replication Efficiency of AKAV Was Temporarily Inhibited in the Established Cell Lines

As N protein is essential for AKAV replication, we hypothesized that the established cell lines that stably expressed AKAV N protein would be able to promote AKAV replication. However, when we evaluated the AKAV growth kinetics in C8H2 and F7E5 cell lines, we were surprised to find that AKAV replicated slower in C8H2 and F7E5 cells than in BHK-21 cells. The multistep growth curves showed that AKAV in C8H2 and F7E5 cells presented significantly lower titers than those in BHK-21 cells at 12 hpi (*p* < 0.001), 24 hpi (*p* < 0.001) and 36 hpi (*p* < 0.05) (Figure 6), while no significant difference was found from 48 to 72 hpi. These data suggest that AKAV replication was temporarily inhibited in the cell lines that constitutively expressed the AKAV N protein.

### 3.4. AKAV mRNAs Expression Is Reduced in the Established Cell Lines

In order to verify the hypothesis that the inhibited viral replication resulted from the reduced mRNA expression, the Gc mRNA in the AKAV-infected cell lines (MOI = 0.01) was assessed by the qRT-PCR method. As a result, Gc mRNA expression in both C8H2 and F7E5 cells was lower than that of BHK-21 cells, with extremely significant differences (*p* < 0.001) at 12, 24 and 36 hpi (Figure 7A). Another qRT-PCR targeting AKAV RdRp mRNA was performed to determine whether it is a general phenomenon that AKAV mRNA expression is reduced in C8H2 and F7E5 cell lines. As expected, a similar result was observed (Figure 7B). These findings explained the reason for the suppressed viral replications at the indicated time points.

### 3.5. The Sensitivity of Sera Detection with Purified AKAV N Protein

SDS-PAGE analysis showed that a single protein band with a molecular weight of around 26 kDa was detected in the purified proteins from both C8H2 and F7E5 cells (Figure 8A). The concentrations of the purified AKAV N proteins were determined to be 434 and 512 µg/mL.

An indirect ELISA was conducted using the purified AKAV-N protein as the coating antigen. As shown in Figure 8B, the average OD_450_ values of the AKAK antiserum decreased as the dilution fold increased from 2^1^ to 2^10^, while the decreasing trend stopped at the dilution of 1:2^11^. The OD_450_ values of the AKAK antiserum diluted from 1:2^11^ to 1:2^15^ were comparable to the levels of the negative control.

## 4. Discussion

A large-scale serological survey of Akabane virus infection in 2017 indicated widespread AKAV infection in China among cattle and sheep [24], but little attention has been paid to it as it deserves. In the past three years, AKAV has frequently struck Jilin Province, raising public concern about the large epidemics of AKAV in China. Therefore, it is of great importance to establish methods that can be applied to the diagnosis of AKAV infection and to have a better understanding of this virus.

As a member of the orthobunyaviruses, AKAV shares most of its characteristics with the other viruses (such as Schmallenberg virus, SBV) of this genus. In terms of the reason that the S-segment is the most conserved genomic region within individual orthobunyavirus species and the N protein it encodes is the most abundant protein in virions and infected cells [10], it is currently widely used for virus or antibody detection [18,19,22,25,26,27]. In the present study, we successfully established and characterized two stable BHK-21 cell lines (C8H2 and F7E5) in which the AKAV N protein is constitutively expressed (Figure 4A). The generated cell lines provide a candidate source for the production of the AKAV N protein (Figure 8A). The result of the indirect ELISA assay (Figure 8B) showed a promising way to detect the AKAV infection by ELISA based on the AKAV N proteins extracted from these cell lines, while further optimization of the ELISA conditions is required. The IFA assay reflected that C8H2 and F7E5 cells can react with the antibody to AKAV directly (Figure 4B). This suggests that the established cell lines could be used as a safe antigen matrix for the IFA detection of anti-AKAV antibodies in serum samples. However, before the indicated ELISA or IFA method is applied, it is important to investigate the cross-reactivity with the antisera of other bunyaviruses within the Simbu serogroup.

Moreover, the stable overexpression of the AKAV N protein in the constructed cell lines enables the recovery of replication-defective AKAV strains. This can be achieved by co-transfecting plasmids carrying the AKAV-L, AKAV-M and AKAV-S△N segments into C8H2 or F7E5 cells. The AKAV virions can then be assembled with the help of the AKAV N protein expressed by the cells. The rescued AKAV strain can only propagate in C8H2 or F7E5 cells and are replication-defective in other cells, making it a safe and promising candidate vaccine for AKAV prevention.

The N protein of all (–)ssRNA viruses has the main function of encapsulating vRNA and cRNA by forming ribonucleoprotein (RNP) complexes, thereby protecting the viral genomes from host defense mechanisms [28]. Except for the primary role of providing structural uniformity to the RNA genome, the N protein of bunyaviruses also interacts with membrane glycoproteins [29,30,31,32,33] and interacts with the viral RdRp to allow access to the RNP during RNA synthesis [9].

Based on the above, we initially hypothesized that the overexpressed AKAV N protein would promote the replication of AKAV. However, the growth kinetics analysis showed that the AKAV N protein-overexpressing cell lines suppressed viral replication rather than facilitating it (Figure 6). Many factors, such as interference with adsorption, gene expression, assembly, release and other processes, can prevent viral replication. The same viral titers in C8H2, F7E5 and BHK-21 cells at 0 hpi (Figure 6) revealed the adsorption process was not hampered. By conducting the qRT-PCR assays, we found that the expression of the Gc mRNA was decreased (Figure 7A). Gc protein, together with Gn protein, plays critical roles in mediating virus assembly, the formation of virus particles, and attachment to new target cells [34]. Therefore, the propagation of AKAV virions is undoubtedly curbed by reduced Gc protein synthesis. Except for the Gc gene on the M segment, the RdRp gene on the L segment was also detected to be reduced in mRNA expression (Figure 7B), which suggested that the overall gene expression of AKAV was downregulated. The aforementioned findings showed that the stably overexpressed AKAV N protein in BHK-21 cells can temporarily suppress viral replication by decreasing viral mRNA expression.

Similar observations were reported in another bunyavirus, Rift Valley Fever Virus (RVFV), in which the expression of the recombinant N protein showed pathogen-specific resistance to RVFV infection [35]. The results of this referred article strongly suggest that it is the N sequence but not the N protein that is responsible for the inhibitory effect [35]. In contrast, another study demonstrated that expression of Borna Disease Virus (BDV) N, P or X proteins rather than a non-translatable viral RNA derived from the N gene rendered human cells resistant to subsequent challenges with BDV [36]. The resistance resulted from the unbalanced expression of viral nucleocapsid components that selectively blocked the polymerase activity of viruses [36]. For bunyaviruses, the formation of the RNP depends on the association of viral RNA with multiple copies of the N protein, and, consequently, the N: RNA interaction is critical for virus viability [9]. Whether there is also a balanced ratio between the AKAV N protein and the RNA or other viral proteins deserves further analysis.

In addition to viral N proteins, trans-acting viral proteins [37,38,39], RNA interferences [40,41], non-specific dsRNAs [42,43], defective interfering particles [44] and viruses [45,46] are able to mediate the resistance of cells to subsequent viral infections. The resistance might be caused by the competition between the co-infected viruses for metabolites, replication sites [47] or viral replication requiring host proteins [48,49]. It is yet unknown if the resistance phenomena in the current investigation may be explained by the same processes. Consequently, the specific mechanism by which the overexpressed N protein in the infected BHK-21 cells leads to decreased AKAV mRNA expression will be the subject of our next line of research.

In conclusion, we developed and characterized two stable BHK-21 cell lines (C8H2 and F7E5) that express the AKAV N protein constitutively. These cell lines may be used to develop the replication-defective AKAV vaccine for AKAV prevention and the ELISA or IFA approach for AKAV detection. Also, we discovered that the stably produced AKAV N protein in BHK-21 cells prevented viral replication by lowering viral mRNA expression, shedding fresh information on the target for AKAV treatment and prevention.

## Figures and Tables

**Figure 1 pathogens-12-01058-f001:**
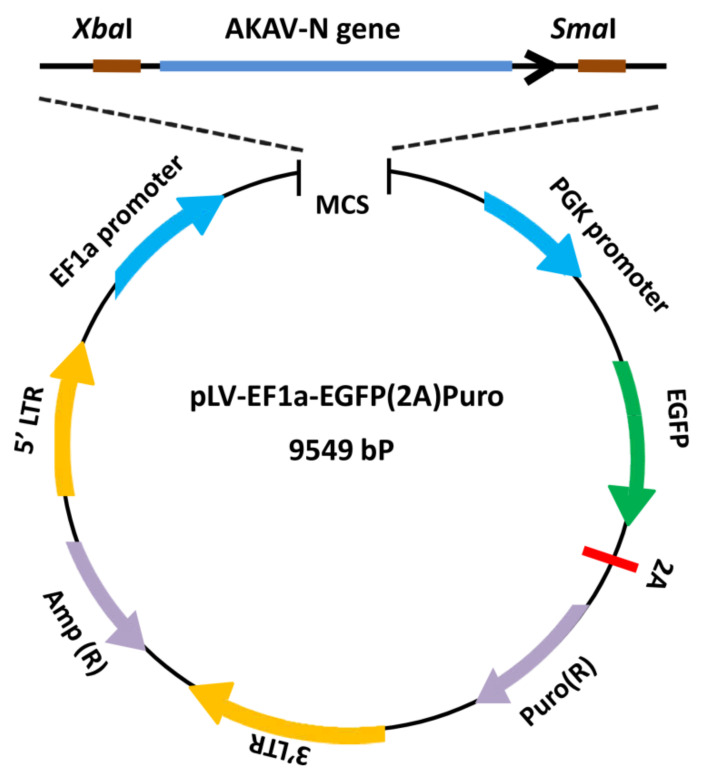
Schematic diagram showing the genome organization of the recombinant plasmid. The N gene of AKAV was inserted into the lentiviral vector pLV-EF1a-EGFP(2A)Puro between *Xba*I and *Sma*I to generate pLV-EGFP-AKAV-N. Cells that underwent recombination can be conveniently monitored by observing EGFP fluorescence and can be selected by puromycin.

**Figure 2 pathogens-12-01058-f002:**
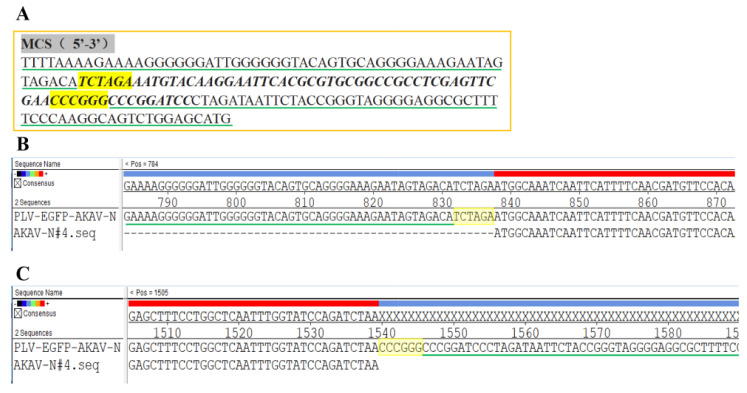
Sequencing identification of pLV-EGFP-AKAV-N. The genome around the recombination region of pLV-EGFP-AKAV-N was sequenced and aligned with the sequences of AKAV-N and pLV-EF1a-EGFP(2A)Puro. (**A**) showed the reference sequence of the MCS region of pLV-EF1a-EGFP(2A)Puro. The italics indicate the sequence of the MCS. The neighboring regions of MCS were underlined in green. The sequences of *Xba*I and *Sma*I were highlighted in yellow. (**B**,**C**) showed partial details of the sequence alignment result. The sequences of *Xba*I and *Sma*I were highlighted in yellow, and the sequences that corresponded with the vector pLV-EF1a-EGFP(2A)Puro were underlined in green.

**Figure 3 pathogens-12-01058-f003:**
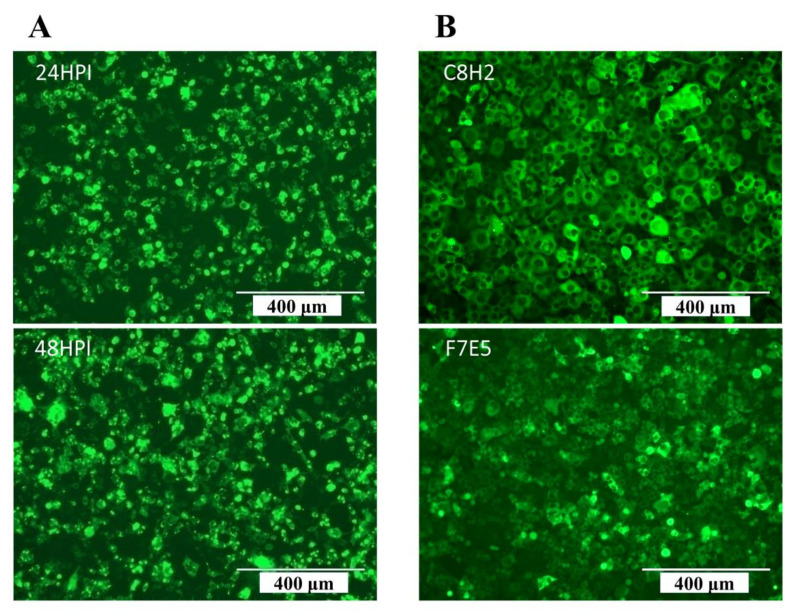
Fluorescent microscopic analysis of the generated cell lines. (**A**) Fluorescence images of 293 T cells posttransfection. (**B**) Fluorescence images of the EGFP-positive BHK-21 cell lines, which were designated as C8H2 and F7E5, respectively. These positive cell lines were selected by puromycin (5 µg/mL) and subcloned by the limiting dilution method. Scale bars indicate 400 µm.

**Figure 4 pathogens-12-01058-f004:**
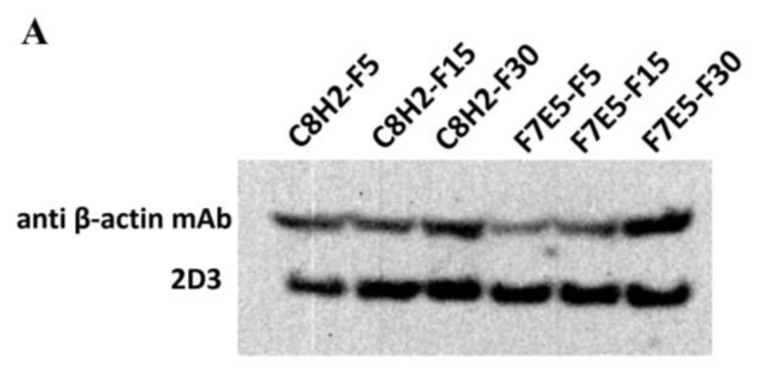
Expression of the AKAV N protein in C8H2 and F7E5 cell lines. (**A**) Western blot analysis of AKAV N protein expression in C8H2 and F7E5 cell lines. Proteins in the lysates of C8H2 and F7E5 cells were separated by SDS-PAGE, followed by identification with Western blot by using monoclonal antibodies against AKAV N (2D3) or β-actin. (**B**) IFA analysis of AKAV N protein expression in C8H2 and F7E5 cell lines. MAb 2D3 was used as the primary antibody and the RBITC-labeled goat anti-mouse IgG was used as the secondary antibody. Green, auto-fluorescence EGFP; red, immunostaining fluorescence corresponding to the AKAV N protein. Scale bars indicate 400 µm.

**Figure 5 pathogens-12-01058-f005:**
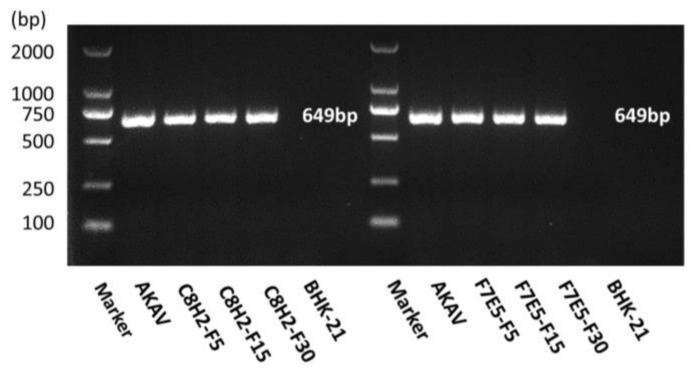
RT-PCR detection of the AKAV N gene in C8H2 and F7E5 cell lines. Total RNA from the 5th, 15th, and 30th generations of C8H2 and F7E5 cells was extracted and transcribed into full-length cDNAs. Then, the cDNAs were used as the amplification template for PCR detection. The cDNA of AKAV was used as the positive control, and the cDNA of BHK-21 cells was used as the negative control.

**Figure 6 pathogens-12-01058-f006:**
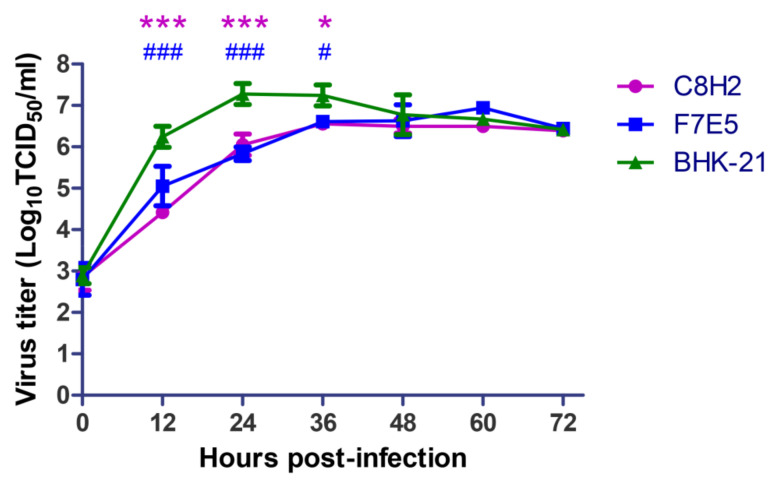
Growth curves of AKAV in BHK-21 cells and the established cell lines. AKAV-infected BHK-21 cells, C8H2 cells or F7E5 cells at an MOI of 0.01. Virus titers were detected by a microtitration infectivity assay in BHK-21 cells. All points represent means ± standard deviations (SD) (error bars) from three independent experiments. Statistical differences are labeled according to a two-way ANOVA followed by a Bonferroni posttest. * indicates a significant difference between C8H2 cells and BHK-21cells (*, *p* < 0.05; ***, *p* < 0.001). # indicates a significant difference between F7E5 cells and BHK-21cells (#, *p* < 0.05; ###, *p* < 0.001).

**Figure 7 pathogens-12-01058-f007:**
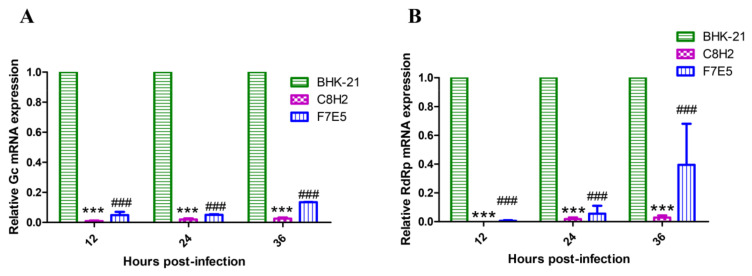
The mRNA levels of AKAV in the infected cell lines. C8H2, F7E5 and BHK-21 cells were infected with AKAV at an MOI of 0.01 for the indicated times. The relative amounts of Gc mRNA (**A**) and RdRp mRNA (**B**) in C8H2 and F7E5 cells were measured by qRT-PCR assay compared to BHK-21 cells. Assays were performed in triplicate, and data were expressed as mean ± SD. Statistical differences are labeled according to a two-way ANOVA followed by a Bonferroni posttest. * indicates a significant difference between C8H2 cells and BHK-21cells (***, *p* < 0.001). # indicates a significant difference between F7E5 cells and BHK-21cells (###, *p* < 0.001). ns, not significant.

**Figure 8 pathogens-12-01058-f008:**
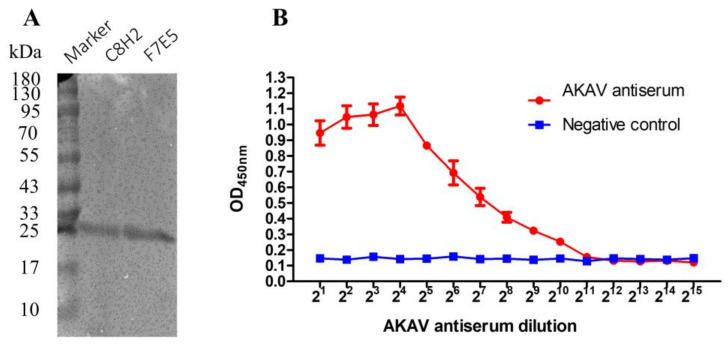
The sensitivity of sera detection with purified AKAV N protein. (**A**) SDS-PAGE analysis of the purified AKAV N protein. (**B**)Indirect ELISA analysis of the sensitivity of eukaryotic AKAV N protein to detect AKAV antiserum.

**Table 1 pathogens-12-01058-t001:** Primers used in the present study.

Primers	Sequences(5′-3′)	Functions
N-F1	GGTTCTAGAATGGCAAATCAATTCATTTTCAACGATGTTCCACAACGGAATGC	Fragment clone
N-R1	TAACCCGGGTTAGATCTGGATACCAAATTGAGCCA	
N-F2	CGATGTTCCACAACGGAATG	N detection (PCR)
N-R2	AAGCTCTAGCTGCAGGTGAG	
Gc-qF	CAGCATAATGAGCAATGCACGG	Gc detection (qRT-PCR)
Gc-qR	CCGTACCTATCGCTAAGCAACC	
RdRp-qF	AGTAGCCTGTGCCAACAAC	RdRp detection(qRT-PCR)
RdRp-qR	GGTACAGATCACGCTGCAT	
GAPDH-qF	CCTTCCGTGTCCCTACTGCCAAC	GADPH detection(qRT-PCR)
GAPDH-qR	GACGCCTGCTTCACCACCTTCT	

## Data Availability

All data associated with this study are included in the paper.

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
