# Peer review of "Generation of Stable Cell Lines Expressing Akabane Virus N Protein and Insight into Its Function in Viral Replication"

_pathogens, 2023, doi:10.3390/pathogens12081058_

Round 1
Reviewer 1 Report
Overall the work by Wang et al, describing Generation of Stable Cell Lines Expressing Akabane Virus N 2 Protein and Insight into Its Function in Viral Replication is interesting but the study needs further experiments to be impactful.
Specific comments:
1. The observation that over expression of N protein inhibits Akabane virus replication warrants further investigation (Figure 6 and 7). The authors have pointed that different viruses employ different strategies and this must be evaluated in Akabane virus.
2. The IF based staining of N protein looks different in two cell lines (excluded from the nucleus in C8H2, while not in other) (Figure 3 and 4). Can the authors explain this?
3. The authors mention in discussion that the cell line can be used for AKAV N protein production and other applications. While this would be useful, further experiments should be done. The authors should show the sensitivity of sera detection, purify AKAV N protein (maybe small scale) and show the functionality.
4. Also the authors should explain the smaller bands in Figure 5.
Minor editing of the language is needed.
Reviewer 2 Report
There are inaccuracies throughout the text, you must pay attention to the correct use of terms: fig.1 (line 91) genome instead of gene, line 123 nucleotide instead of Nucleocapsid and fig. 7 (line 263) MRNA instead of mRNA
Line 142-154: qRT-PCR Assay. After DNAsi treatment how do you check for the residual genomic contamination? DNA escaped from DNAsi treatment would create problems in the quantification of mRNA level , to be sure not to detect DNA you should insert proper controls
Sometimes is difficult to follow a logical thread, results of some experiments are not described in “Materials and Methods”. Line 219: Validation of the stability of AKAV N gene. PCR protocol used is not described in “Materials and Methods”, you cite it only in “Results”. It is improper to evaluate gene expression using PCR; your results demonstrate the integration of the N gene but not the N protein expression, the sentence in line 224-226 is not correct. You checked for the N protein expression by WB but is not clear at which time points, for this reason you do not demonstrate the stable expression of the N protein. You wrote in the paper (line 242 and line 260) that AKAV replication is inhibited in C8H2 and F7E5 cell lines given that mRNA expression decrease compare to BHK21, how can you say in line 225-226 that you have been established a cell line consistently expressed N protein?
Review the English form of the paper with the help of a native speaker
Round 2
Reviewer 1 Report
The authors have addressed my concerns. With language editing the manuscript can be accepted.
Minor editing of the language needed
